# Design of a Horizontal Axis Tidal Turbine for Less Energetic Current Velocity Profiles

**Job Immanuel Encarnacion [1,2,*]**, **Cameron Johnstone [2]** and **Stephanie Ordonez-Sanchez [2]**

1    Department of Mechanical Engineering, University of the Philippines, Quezon City 1101, Philippines
2    Energy Systems Research Unit, University of Strathclyde, Glasgow G1 1XJ, UK
*    Correspondence: job.encarnacion@strath.ac.uk or jbencarnacion1@up.edu.ph; Tel.: +44-7380-299912

**Abstract:** Existing installations of tidal-stream turbines are undertaken in energetic sites with flow speeds greater than 2 m/s. Sites with lower velocities will produce far less power and may not be as economically viable when using "conventional" tidal turbine designs. However, designing turbines for these less energetic conditions may improve the global viability of tidal technology. Lower hydrodynamic loads are expected, allowing for cost reduction through downsizing and using cheaper materials. This work presents a design methodology for low-solidity high tip-speed ratio turbines aimed to operate at less energetic flows with velocities less than 1.5 m/s. Turbines operating under representative real-site conditions in Mexico and the Philippines are evaluated using a quasi-unsteady blade element momentum method. Blade geometry alterations are undertaken using a scaling factor applied to chord and twist distributions. A parametric filtering and multi-objective decision model is used to select the optimum design among the generated blade variations. It was found that the low-solidity high tip-speed ratio blades lead to a slight power drop of less than 8.5% when compared to the "conventional" blade geometries. Nonetheless, an increase in rotational speed, reaching a tip-speed ratio (TSR) of 7.75, combined with huge reduction in the torque requirement of as much as 30% paves the way for reduced costs from generator downsizing and simplified power take-off mechanisms.

**Keywords:** blade element momentum; solidity; tip-speed ratio; tidal turbine; low velocity; less energetic current; design optimisation

## 1. Introduction

The periodic rise and fall of sea levels are attributed to the gravitational interaction between the sun and moon combined with the rotation of the earth. This motion carries a predictable and reliable source of energy since the relative motion of these bodies occurs at an astronomical time scale. Large-scale tidal power plants that close off a portion of the shore and act as sea-based dams have proven to be effective in converting this energy into usable power. These dams incur large capital costs as very large and sturdy foundations are required in closing off any part of the sea or bay.

Tidal stream turbines (TSTs) provide an alternative in harnessing energy from marine currents as they can be used in the open sea without the need to enclose large areas of maritime space. However, these devices are still more expensive [1] than other forms of renewable technology. Horizontal axis tidal turbines (HATTs) are currently the dominant device type [2], with multiple developers in the pre-commercial and commercial phase of implementation.

As of the time of writing, Scotrenewables SR2000 [3] deployed in Orkney, UK, is the world's most powerful turbine operating on a full commercial scale. Two 16 m tidal turbines with a shared floating platform are capable of producing 2 MW at a rated speed of 3 m/s. Another project in Orkney, Atlantis Meygen [4], is one of the largest planned full commercial scale tidal turbine array with four

AR1500 single-rotor turbines, each capable of producing 1.5 MW. Atlantis is also set to test the AR2000, a single 20–24 m diameter turbine, which should be capable of producing 2 MW at speeds of more than 3 m/s [5]. The Nova M100 [6] is a smaller 9 m turbine with a rated capacity of 100 kW at a rated speed of 2 m/s. Three M100 turbines are deployed in the world's first fully-operational grid-connected tidal turbine array in Shetland, UK.

Outside the UK, the Sabella D10 [7] is a 10 m diameter 6-bladed turbine capable of producing 1 MW at a speed of 4 m/s. The device is set to be the first marine current turbine that will provide electricity to the French energy network. The Verdant Power Roosevelt Island Tidal Energy [8] project in the US is a tidal array project aiming to deploy thirty Verdant Gen5 5 m diameter turbines with capacities of 35 kW at a speed of 2.5 m/s. Back in 2006, Verdant successfully demonstrated the operation of a grid-connected array with six of their Gen4 full-scale turbines, with 9000 turbine-hours of operation [9].

While success has been achieved in commercialisation, the technology generally faces the challenge of being highly site-specific as existing designs are geared toward current velocities greater than 2 m/s [10,11], while most of the world's oceans have velocities less than this value. Less energetic flows have less than half of the extractable power from most northern UK sites ($P \propto V^3$). Nonetheless, there is a considerable increase in the number of exploitable sites, while lower structural loading resulting from a less energetic flow should translate to lower cost of manufacturing, materials, operation, and maintenance.

The south, west, and east coasts of the UK are included in the less energetic current category, with current velocities of less than 2 m/s [12,13]. Deployment of tidal energy devices are planned to harness energy in the tidal channel of Ria Formosa, Portugal, although the same challenge is faced as the site is limited to a current velocity of 1.4 m/s. Initial deployment tests were performed showing underutilisation of the small-scale Evopod turbine [14,15], with a maximum output below 25% of the rated power output [16].

Less energetic flows dominate South East Asia [17], with current velocities less than 2 m/s. The San Bernardino Strait in the Philippines has been identified as a potential tidal energy site, with current velocities reaching upto 4.5 m/s [18] near the southern tip of the Capul islands. However, the country is still characterised by flow speeds of less than 2 m/s, with most areas reaching current velocities of 1.4 m/s [19,20]. Malaysia has current velocities reaching up to 1.2 m/s [21], which limits the possible operation of current TST designs to near cut-in speeds, leaving the turbine underutilised. Other tropical countries such as Brazil and Mexico also have similar flow conditions. Brazil has slightly better conditions, with peak current velocities reaching a range of 2–2.5 m/s, although median speeds still remain at 1.1 m/s [22].

The Yucatan channel in Mexico is a passageway that has current velocities with an upper bound of 2 m/s [23,24], but much of the passing current is limited to below 1.4 m/s [25]. Nonetheless, continuous operation is possible since the Yucatan current is a constant marine current as opposed to the periodic tidal current present in the previously mentioned sites. A similar marine current is found near Taiwan, with an average flow speed of less than 1.5 m/s [26]. The Kuroshio current has been identified as a potential tidal energy resource for the country, leading to the research and development of the floating Kuroshio turbines [27–29].

Reduction in capital cost is needed to make these sites more viable for tidal technology, and may be achieved through downsizing and utilisation of cheaper materials. Additional ways to reduce cost are possible and involves designing TSTs that would have an optimal operation under a less energetic flow, with velocities ranging from 1 m/s to 1.5 m/s. TST rotor designs will typically have a low tip-speed ratio (TSR) operating range, where maximum power is delivered at lower values (TSR < 4), leading to large torque production.

Currently, researchers and developers in Taiwan have been working on a floating turbine that can harness energy from the Kuroshio. Towing tank tests have shown success with a peak generation of over 400 W for a 2 m diameter turbine [29]. However, the design still achieves maximum output at low

TSRs (TSR $\approx$ 4). This will require large generators and complex power take-off mechanisms to bring up the rotational speed. The new design methodology hypothesises that enabling operation at higher TSRs (TSR > 6) in less energetic flow conditions would lead to savings since it would require smaller generators due to reduced torque requirements.

Large torque production is not possible in less energetic flows, but the amount of power output may be comparable by utilising higher angular speeds (Power = torque × angular speed). Direct-drive or at least a less complex power take-off mechanism could be employed with a faster rotational speed. This would allow for further reductions in cost. Smaller diameter rotors with $\phi < 10$ m provide a benefit of further increasing rotational speed in addition to reducing capital cost.

A design methodology for horizontal axis tidal turbines operating in less energetic currents is presented. The study outlines the design process and its application to two different aerofoil-specific blade geometries that operate at different TSR ranges. A steady-state performance filtering is undertaken to filter through different blade variations before evaluating the performance of the optimised blade geometry operating under a velocity profile obtained from two sites in Asia and America.

The analysis of the resulting blade geometries serves as an investigation of the hypothesised benefits of designing blades that operate at higher TSRs in less energetic currents according to (a) lower torque production at a (b) reasonable decrease in power output. The study also investigates how far the TSR location of the maximum $C_P$ may be pushed towards higher values as a result of the blade alterations.

## 2. Methods

This section presents the methodology used to alter the blade geometry and the numerical set-up used to evaluate the performance and loads on the turbine rotor. The steady-state evaluation of the different blade geometries served as a quick filter to determine which blades are deemed to be feasible or optimal under the governing flow conditions. Profiled flow from surveyed data was utilised to evaluate the performance of the optimised blade under real-site conditions.

### *2.1. Blade Modifications*

#### 2.1.1. Distribution Modifications

Two aerofoil-specific blade geometries are analysed in this study: a NACA 638xx blade ($TSR_{Cp\text{max}} \approx 5.75$) [30] and an NREL S814 blade ($TSR_{Cp\text{max}} \approx 4$) designed for low-Reynolds-number flow [31]. Comparison of the performance of the blades with the subsequent parametric modifications gives an outlook on the challenges and benefits of using a high-TSR blade in a less energetic environment, as opposed to using a usual low-TSR blade.

Each blade geometry from [30,31] was altered using two parameters: (a) chord distribution and (b) twist distribution. The published chord and twist values for the root and tip sections of [30,31] were used as references for the application of scaling coefficients given by the conic equation:

$$r^2 + \mathbf{A}\lambda r + \lambda^2 = 1, \tag{1}$$

where $\mathbf{A}$ determines the shape of the distributions defined by the radial coordinates, $r$, of the control section and the corresponding scaling coefficients, $\lambda$. The values of $\lambda$ always fall within $0 \le \lambda < 1$ since $r$ is limited to $0 < r \le 1$. Altering the values of $\mathbf{A}$ results in either an elliptical ($\mathbf{A} < 2$), linear ($\mathbf{A} = 2$), or hyperbolic ($\mathbf{A} > 2$) graphical pattern. The graphical patterns alter the chord and twist values while

still maintaining a smooth change in distribution along the blade span. These distributions serve as an initial case study on their performance under less energetic flow. $\lambda(r)$ is then applied as follows:

$$y_{new}(r) = y_{tip} + y^* * \lambda^*(r), \tag{2}$$

$$y^* = y_{root,base} - y_{tip}, \tag{3}$$

$$\lambda^*(r) = \lambda(r)/\lambda_{root}, \tag{4}$$

where $y$ represents the chord and twist values along the blade span, set as a function of $r$. Using $y^*$ and $\lambda^*$ ensure that the chord and twist values at the root and tip remain constant in all blade variations, making it possible to isolate the effect of how the distribution affects performance independent of the absolute difference between the root and tip sections.

Notice that $y^*$ is not a function of $r$ and is only the difference between the root and tip values of each distribution. This ensures that all resulting $y$ values are scaled with the $x$ axis ($y = 0$) as reference. Utilising $y^*$ instead of applying each coefficient to each old $y$ value limits the distributions to the predetermined graphical patterns; not all base case distributions are linear (see Figures 1 and 2). Equations (2)–(4) may be modified to allow for other design alterations, such as a more aggressive initial decrease in $y$ values, and may be considered for further study.

Figures 1 and 2 show the scaling applied to both blade geometries. Elliptical distributions were produced with **A** = 0.5, while hyperbolic distributions were produced with **A** = 5. Base chord distributions for both geometries follow a linear trend while the base twist distributions are generally hyperbolic.

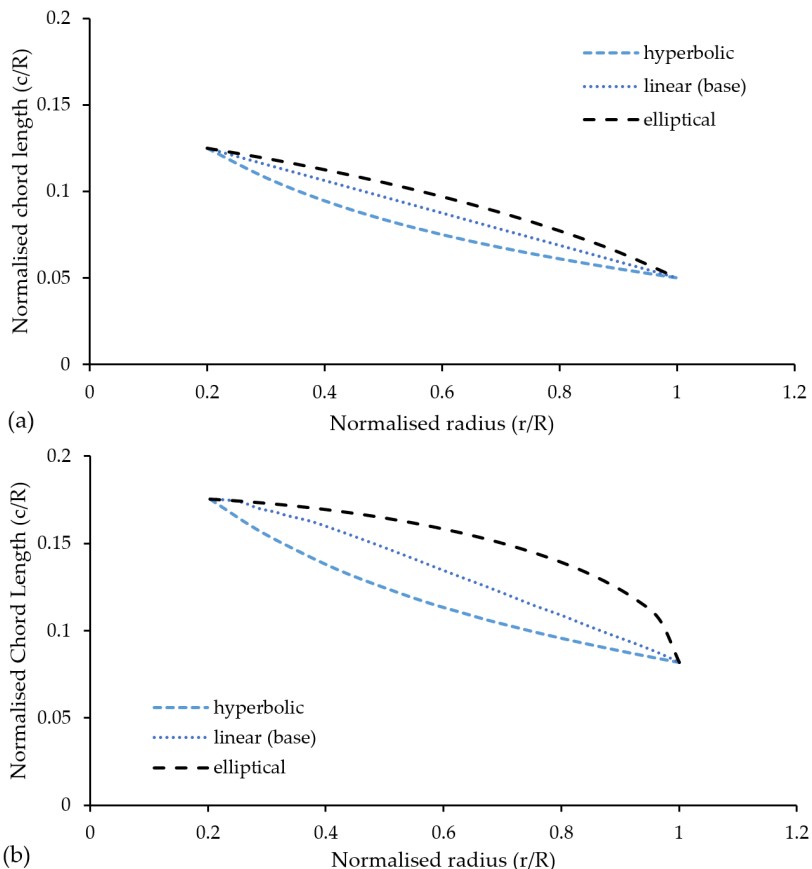

**Figure 1.** Normalised chord length distributions for the (**a**) NACA and (**b**) NREL blades.

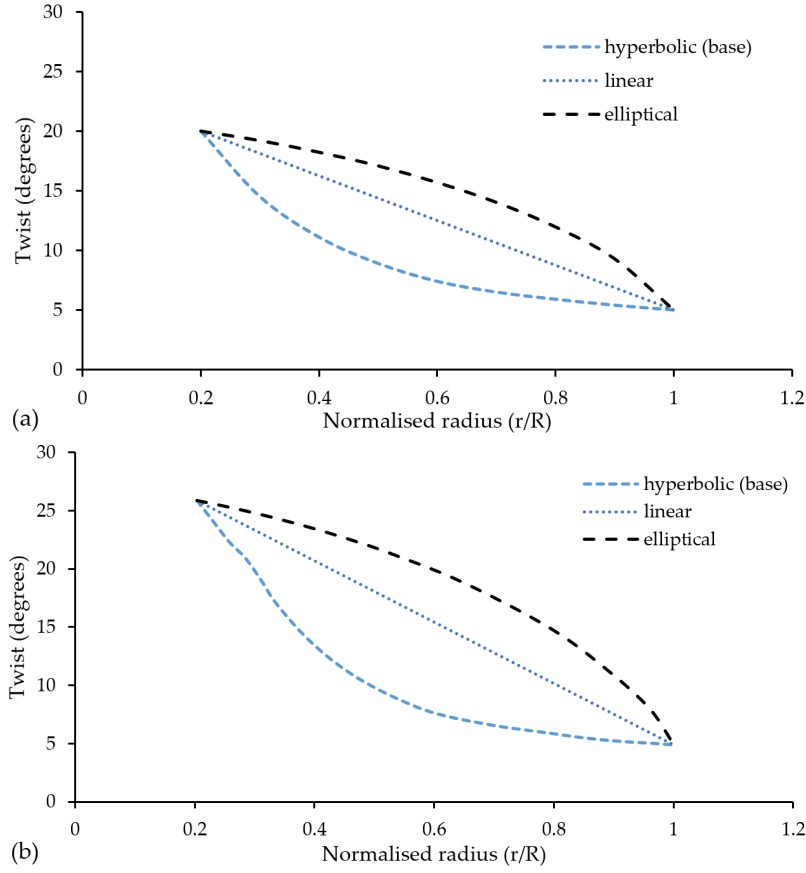

**Figure 2.** Twist distributions for (**a**) NACA, pitch = 0 deg, and (**b**) NREL, pitch = 1.75 deg.

### 2.1.2. High-TSR Blade Modification

A high-TSR blade may be achieved making the blades more slender [32]. This feature produces less drag as an effect of the lower solidity resulting from the chord length reduction. The original NACA blade geometry already has a more slender profile than the original NREL blade geometry. Isolating the effect of reducing the solidity of the blades is done by subjecting both blade geometries to chord length reduction while maintaining the set number of blades and same graphical patterns. This allows for a comparison of the original version and the low-solidity version of the aerofoil-specific blade geometry.

An iterative process was adopted to identify the most effective reduction to be applied to the blade sections. This resulted in a scaled reduction of 37.5% of the tip chord length being adopted to maintain the elliptical/linear/hyperbolic trends. Tables 1 and 2 show the chord distribution of the low-solidity blades.

**Table 1.** Blade geometry specifications for low-solidity NACA blades (37.5% reduced chord length).

| Normalised Radius (r/R) | Normalised Chord Length (c/R) | | |
|:---:|:---:|:---:|:---:|
| | **Hyperbolic** | **Base** | **Elliptical** |
| 0.2 | 0.106 | 0.106 | 0.106 |
| 0.3 | 0.0881 | 0.097 | 0.102 |
| 0.4 | 0.075 | 0.088 | 0.097 |
| 0.5 | 0.0647 | 0.078 | 0.092 |
| 0.6 | 0.056 | 0.069 | 0.085 |
| 0.7 | 0.049 | 0.059 | 0.076 |
| 0.8 | 0.043 | 0.05 | 0.066 |
| 0.9 | 0.036 | 0.041 | 0.053 |
| 1 | 0.031 | 0.031 | 0.031 |

**Table 2.** Blade geometry specifications for low-solidity NREL blades (37.5% reduced chord length).

| Normalised Radius (r/R) | Normalised Chord Length (c/R) | | |
|:---:|:---:|:---:|:---:|
| | Hyperbolic | Base | Elliptical |
| 0.204 | 0.145 | 0.145 | 0.145 |
| 0.232 | 0.138 | 0.144 | 0.144 |
| 0.259 | 0.132 | 0.143 | 0.143 |
| 0.285 | 0.127 | 0.139 | 0.143 |
| 0.313 | 0.122 | 0.137 | 0.142 |
| 0.334 | 0.118 | 0.135 | 0.141 |
| 0.376 | 0.111 | 0.132 | 0.140 |
| 0.417 | 0.105 | 0.127 | 0.138 |
| 0.460 | 0.099 | 0.122 | 0.136 |
| 0.502 | 0.094 | 0.117 | 0.134 |
| 0.544 | 0.089 | 0.111 | 0.131 |
| 0.587 | 0.084 | 0.105 | 0.128 |
| 0.630 | 0.080 | 0.100 | 0.125 |
| 0.672 | 0.076 | 0.095 | 0.122 |
| 0.713 | 0.072 | 0.089 | 0.118 |
| 0.755 | 0.069 | 0.084 | 0.114 |
| 0.799 | 0.065 | 0.078 | 0.109 |
| 0.839 | 0.062 | 0.073 | 0.103 |
| 0.884 | 0.059 | 0.067 | 0.096 |
| 0.926 | 0.056 | 0.062 | 0.087 |
| 0.967 | 0.053 | 0.056 | 0.076 |
| 1.000 | 0.051 | 0.051 | 0.051 |

## 2.2. Analytical Model

### 2.2.1. Steady-State Characterisation

Blade element momentum (BEM) theory was used to evaluate the hydrodynamic performance of the turbine rotor in terms of the coefficient of power, $C_P$, and the coefficient of thrust, $C_T$. This method is used extensively in wind and marine energy engineering since it does not require the massive computational overhead needed for CFD analysis. Excellent agreement between CFD and BEM performance characterisations have been observed by Bangga [33] for wind turbines and Faudot [34] for tidal turbines. Unsteady BEM for tidal turbines was also found to have a similar predictive accuracy to transient CFD simulations [35].

The BEM model in this study follows the solution method of Ning [36], employing convergence based on the inflow angle $\phi$ instead of the axial and tangential induction factors $a$ and $a'$, respectively. This approach allows for the utilisation of a root-finding solution method, which is faster than error minimisation with two parameters. Loads and performance are still obtained using $a$ and $a'$ with power calculated directly from $C_P$ or the combination of torque and rotational speed.

The method starts similarly to the conventional two-parameter method, until the axial and tangential force coefficients are calculated. The procedural change begins as the induction factors are recomputed. This starts the set-up for the one-equation root finding problem. Two convenience variables, $\kappa$ and $\kappa'$, are taken as functions of $\phi$ and used to solve $a$ and $a'$, respectively (Equations (5) and (6)).

$$\kappa(\phi) = \frac{\sigma' c_a(\phi)}{4F(\phi)sin^2\phi} \qquad \kappa'(\phi) = \frac{\sigma' c_a(\phi)}{4F(\phi)sin\phi \cos \phi}, \tag{5}$$

$$a(\phi) = \frac{\kappa(\phi)}{1 + \kappa(\phi)} \qquad a'(\phi) = \frac{\kappa'(\phi)}{1 - \kappa'(\phi)}, \tag{6}$$

where $c_a$ is the axial force coefficient, $\sigma'$ is the local solidity, and $F$ is the tip and hub correction factor used.

Since both induction factors are set as functions of $\phi$, it is then possible to get a single equation for the residual function (Equation (8)).

$$tan\phi = \frac{1-a}{(1+a')\lambda_r},\tag{7}$$

$$f(\phi) = \frac{sin\phi}{1-a} - \frac{cos\phi}{\lambda_r(1+a')} = 0,\tag{8}$$

The presented residual function applies to most of the calculations but will yield invalid results for regimes outside the momentum region. A slightly different form is presented by [36], but it is noted that this would seldom be the case. A full derivation is available from [36], which includes the initialisation of $a$ and $a'$.

Limitations in BEM theory were considered by applying several correction factors to each blade element iteration. The Prandtl tip and hub correction factors were used to account for non-ideal flow from the root and tip otherwise not captured since BEM treats each element as independent, idealised 2D sections [37]. Axial induction factors greater than the theoretical limit, making the momentum theory invalid, were adjusted using the Buhl high induction correction factor [38]. Finally, the Viterna–Corrigan model [39] was used to obtain the aerodynamics of each aerofoil upon the onset of stall.

Aerodynamic characteristics (**Re** = 500,000) of the aerofoil sections were evaluated through ANSYS Fluent, which provides a quick aerodynamic characterisation for numerous aerofoils (NACA 638xx family), at angles of attack of $-20°$ to $16°$. A structured quadrilateral-element C-grid was used to mesh the fluid domain for each aerofoil section. Domain edges were set to be 20 chord lengths away from the surface of the foil. The elements were configured to have a minimum orthogonality of 0.7 and maximum skewness of 0.35 from within 30% chord length of the aerofoil's surface. The mesh was refined near the aerofoil surface, with at least 15 elements within the boundary layer and the first cell height set at $5 \times 10^{-5}$ m. The $k$-$\omega$ SST model was used to resolve the RANS equations as the flow separated at most positive angles of attack. It was determined that approximately 52,000 cells were sufficiently accurate, as doubling the cell quantity gave marginal change in the coefficients at roughly 1% difference. The lift and drag coefficients were then accessed as tabular data exported from ANSYS Fluent results.

2.2.2. Quasi-Unsteady BEM for Profiled Flows

Steady-state characterisation assumes uniform inflow across the whole turbine, which is not the case for real sites. A depth-dependent current velocity function, $U(z)$, was used to approximate the velocity profile applied on the turbine rotor during operation. $U(z)$ can be defined through a power law function [40]:

$$U(z) = U_o z^{1/b},\tag{9}$$

where $U_o$ is the tidal velocity at the surface, $z$ is the distance from the bottom of the seabed, and $b$ is theoretically equal to 7, which might not be applicable to less energetic environments. Real-site data, detailed in Section 2.3, is fitted to get approximate values for $U_o$ and $b$, according to each site. These values of $U_o$ and $b$ are constant with respect to time, which means an unchanging velocity profile $U(z)$. The unsteady characterisation then pertains to the change in inflow conditions for each blade as it goes through the profiled flow rather than a changing inflow over the whole rotor. Thus, power output is expected to be roughly constant over the whole time simulation.

The velocity function was applied element-wise [41] on a 4 m diameter turbine positioned at the middle of the water column (Figure 3a), operating at optimum TSR. A quasi-unsteady simulation

was done by assuming each time step as a steady-state snapshot of performance. Load variations result from the vertical displacements of the blade elements along the water column. The azimuthal angle, $\Phi_{B,i}$, where $i$ denotes the $i$th blade being analysed, dictates the angular displacement of the blade, which can be transformed to the get the vertical position of any element along the water column. Figure 3b shows the azimuthal coordinate system used herein [41].

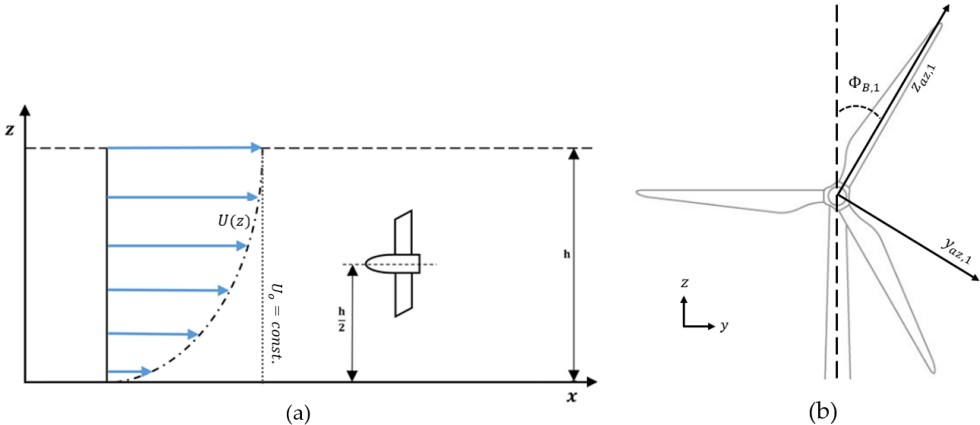

**Figure 3.** Turbine setup with (**a**) global coordinates (**b**) and azimuthal coordinate systems.

### 2.3. Site Cases

Some sites follow the power law with a $b$-value approximately equal to 7 [42]. However, this may not be representative of less energetic environments, in addition to having no universally accepted value for $b$. Thus, ADCP measurements from two sites of interest for tidal turbine development were obtained to provide a better approximation of the performance of a turbine operating under less energetic flows. The ADCP data was fitted to Equation 1 to obtain the values of $U_o$ and $b$, as opposed to utilising the theoretical characterisation of $b = 7$.

#### 2.3.1. Philippines

The energy mix of the Philippines is still dominated by conventional coal-fired power plants, with the share of renewable energy sitting at 24.2% despite the goals set in past years [43]. Additional targets set for 2025 include 75 MW of ocean energy from tidal, wave, and ocean thermal power plants [44].

It is estimated that the tidal in-stream potential of the country is 80 GW [45]. However, most of the sites that carry this potential have less energetic flows and cannot be captured with conventional TSTs designed to operate at current velocities less than 2 m/s. The San Bernardino Strait has been identified as a potential site for tidal in-stream energy, with peak flows of about 4.5 m/s [18], although the average annual current is only between 1.20–1.60 m/s [20]. Other sites in the Philippines have even less energy with annual average velocity magnitudes between 0.40–0.80 m/s [19].

Time-series velocity data recorded in 2015 and provided by a Filipino developer were filtered to obtain the characteristic profile for a typical spring tide. ADCP measurements do not discriminate between tidal current and wave-induced velocities, thus distorting the overall velocity profile [46–49] (Figure 4). The time-averaged velocity profile was then used to analyse the performance of the turbine with the assumption that the average profile is the characteristic profile with a current-only flow condition; the concurrent and counter-current wave-induced effects cancel each other out when the time-series is averaged.

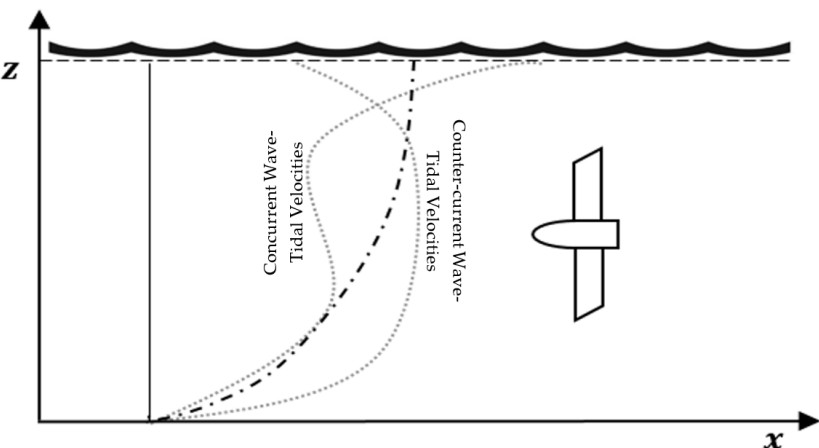

**Figure 4.** Velocity profile distortion due to wave–current interaction.

### 2.3.2. Mexico

The relatively low energy output from less energetic currents can greatly improve how small-to-medium coastal communities attain sufficient power output to meet local demand. Cozumel Island in the east of Mexico has an annual electricity consumption of 274.75 GWh [50] and is projected to grow along with the increase in population and high economic activity dominated by tourism [51].

The Yucatan current passes through the Cozumel Channel, situated between the island of Cozumel and the Yucatan Peninsula [24]. Average velocities within the Cozumel channel may reach up to 2.5 m/s, with speeds of 1.1 m/s taken at 30 m within the water column [23,24]. This makes the adoption of marine renewable energy using TSTs a good alternative for a reliable and sustainable energy supply, which is currently being supplied from the mainland through a single submarine cable.

The current in Cozumel is thermally derived and mainly driven by the constantly flowing Yucatan current. This differs from the Philippines case, where periodic increases and decreases of tidal velocities are observed. ADCP measurements obtained from a spatial variability analysis [25] provide snapshots of the flow within the current as opposed to being time-series measurements. Three 10-m windows (Figure 5) with relatively flat bathymetry were selected for analysis. Values for $U_o$ and $b$ were averaged for each window to ensure that the effect of any extreme variability in the data was minimised. All windows have at least 50 m centre-to-centre spacing to allow for at least 10D spacing [52]. The performance of the turbine was evaluated with reference to the average value of $U_o$ and $b$ for each area.

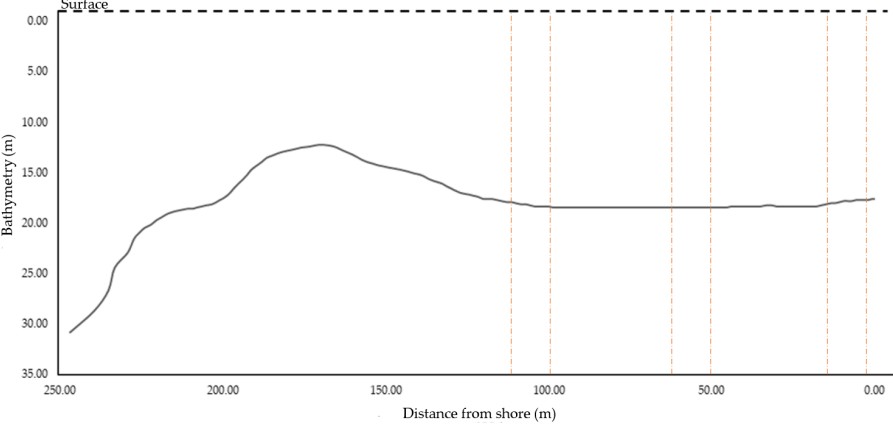

**Figure 5.** Locations of 10-m windows (orange regions) in a section of the Cozumel Channel used for analysis.

*2.4. Design Process*

Two rounds of blade design filtering and selection were undertaken to obtain an optimal blade. Each round reduces the number of blade variations according to specific objectives. These rounds are necessary to minimise the number of simulations needed to produce an optimal design. After the two rounds, each blade variation was then compared to their unaltered forms to evaluate the respective gains in adopting the blade alterations. Figure 6 provides a flowchart of the blade design process used in this study.

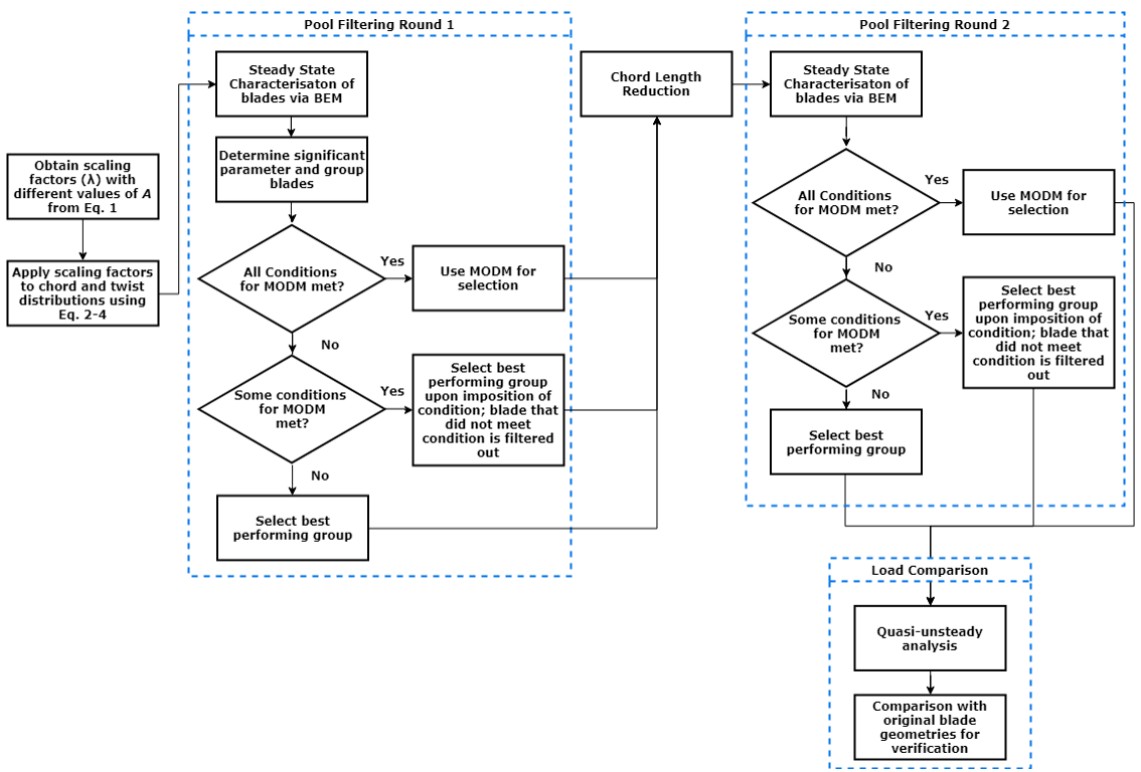

**Figure 6.** Design process for this study. Additional steps may be added to optimise designs.

2.4.1. Pool Filtering (Round 1 and 2)—Steady-State Optimisation

Section 2.1.1 resulted in nine different versions for each aerofoil-specific blade geometry. Steady-state analysis was first adopted to get a blade that would give the best performance. Optimal performance under real-site conditions was expected to follow from the best-performing blade under steady-state flow as the angle of attack in each blade element was optimised non-dimensionally with respect to $C_P$. Candidate blades were then selected according to the following objectives:

1.  Maximum $C_P$
2.  TSR location of maximum $C_P$; higher is better

Thrust was not considered a primary criterion since it was expected to have lower variation compared to torque ($C_T \propto V^2, C_P \propto V^3$). However, thrust would subsequently be considered if the first two objectives resulted in multiple blade profiles and the decision model could not reconcile between multiple equally favourable blades.

**Decision Model**

A multi-objective decision model (MODM) was used to evaluate the blades if no single alteration resulted in the best performance according to the set objectives. The utilisation of MODM allows for the evaluation of blades while considering a greater number of objectives. In this stage, however, a

simplified approach to the MODM was used, with only two primary objectives being evaluated and pool filtering minimising the number of possible *solutions*.

The simplified MODM approach in this study is derived from the known-weights scenario [53], and both objectives are set to have equal weight in the decision process. The weights are applied to scalarised values of the objectives and evaluated using a weighted matrix. Each round also serves as a simplification for MODM since only one parameter is considered per round.

Although maximum $C_P$ and its corresponding TSR location were both considered as primary objectives, this does not mean that a very low maximum $C_P$ occurring at very high TSR, e.g., $C_P = 0.1, TSR = 11$, would be considered as an optimal blade. The evaluation of blade variations according to MODM are then subject to the following conditions, in this order:

1.  The maximum $C_P$ has a value of at least $C_P > 0.35$.
2.  The TSR location of the maximum $C_P$ is reasonably high, with a value of at least $TSR_{Cp\max} > 5.5$.
3.  The difference in maximum $C_P$ between two blade alterations in a specific round is no more than 10.
4.  The difference between the TSR location of the maximum $C_P$ is no more than 1.5 times the TSR location of the lower-TSR blade.

If a condition is not satisfied, the blade that has the best value upon the application of the condition is chosen to be the better among the variations.

These conditions were adopted as initial decision points for the blade design methodology. The conditions minimise the number of simulations, including MODM simulations, needed to select an optimum blade configuration. It is possible that the given thresholds on each condition change as more blade variations are produced and analysed.

**Round 1**

Any difference in performance is attributed to the combined effects of the twist and chord distribution. Nonetheless, it is expected that one parameter would affect rotor performance more than the other [54]. The purpose of round 1 is to identify the parameter that would have a more significant effect on performance according to the set objectives. The blades are then filtered according to the parameter that would give the best performance.

The MODM may be used in this round to evaluate the best blades. However, since only a single parameter is considered, blades are grouped according to the more significant parameter, and the maximum $C_P$ and corresponding TSR locations are averaged within each group. Only the averaged values of the group may be used for MODM.

It is possible that exaggerated values of **A** for any of the two parameters may result in better performance than any of the cases analysed in this study. These exaggerated cases are not part of this study; evaluation of these cases may be performed in a later study where multiple blade variations by **A** may be generated programmatically.

**Round 2**

The filtered blades in round 1 are subjected to further analysis. As described in Section 2.1.2, each blade is subjected to chord length reduction to encourage optimal performance towards higher TSRs. The purpose of round 2 is to select one candidate blade for each aerofoil-specific blade geometry.

In terms of programme optimisation, it is possible to add more blade variations by altering the parameter that was not used for filtering in round 1. In other words, round 2 may be considered as a fine tuning using a single parameter. Nonetheless, only a subset of the nine blade variations is analysed in this study.

2.4.2. Load Comparison—Quasi-Unsteady Loads

Load variation can only be observed upon assumption of a sheared or profiled flow, $U(z)$. The steady-state analyses in the first two rounds assume uniform flow, dictated by $U_o$. This assumption implies that all forces and power drop upon using a profiled flow (Figure 3a). A load comparison was performed between aerofoil-specific blade geometries to substantiate the case for low-solidity high-TSR blades operating in less energetic currents. In particular, the following were evaluated:

1.  the sensitivity or relative drop in load and power upon adoption of a profiled flow from a steady uniform flow,
2.  load cycles and variation in individual blade loads that will result in fatigue.

The load comparison was done for the optimised low-solidity high-TSR blade from round 2 and the original aerofoil-specific blade geometry. A comparison against the original blade profiles was done to quantify the expected performance difference of an optimised high-TSR blade.

## 3. Results

### 3.1. Velocity Profile

The obtained site-specific, curve-fitted values for Equation (1) are shown in Table 3. The values of $b$ for both were less than 7, which indicates a thicker boundary layer. The low flow velocity results in a Reynolds number that is lower compared to sites with an energetic flow, leading to an increase in the boundary layer thickness [55]. However, it may be possible that the sites have a rougher seabed, the influence and effects of which are compounded by low flow velocity.

The thicker boundary layer subjects the rotor to greater velocity variation as the flow decays faster to zero towards the bottom. This means that any turbine deployed in these sites will be operating within a higher velocity shear environment [56].

**Table 3.** Summary of fit for power law (Equation (1)) $U(z) = U_o z^{1/b}$.

| Variable | Case 1: Philippines | Case 2: Mexico [1] |
|---|---|---|
| $U_o$ (surface velocity, m/s) | 0.64 | 1.136 |
| $b$ | 3.5 | 2.991 |
| $h$ (depth, m) | 20 | 18.5 |

[1] Values obtained for Equation (1) are fitted using snapshot values. Any large fluctuation that occurred at the moment of data-capture is not filtered out. Using averaged values for one window should minimise the effects of large fluctuations, but this is not ensured.

### 3.2. Pool Round 1—Steady-State Two-Parameter Sensitivity

The blade alterations resulted in three performance clusters (Figure 7) for each of the aerofoil-specific blade geometries. The clusters are determined by the twist distribution, indicating that the effect of altering the twist distribution has a more significant effect on the performance of a blade compared to the effects of altering the chord distribution. This is because any change in lift and drag coefficients impacts the performance of a turbine significantly [57]. Alteration of the twist distribution directly changes the distribution of the angle of attacks, subsequently changing the effective lift and drag on each blade element.

Filtering was then done according to the performance dictated by the twist distribution. The averaged values for the maximum $C_P$ and the corresponding TSR location are shown in Table 4. MODM was not utilised for round 1 because the smallest difference between the averaged maximum $C_P$ was calculated at 12.2%, which violates condition 1 for MODM (diff$_{C_P\text{max}}$ > 10%).

The base twist distribution resulted in the highest averaged maximum $C_P$ in both aerofoil-specific blade geometries. This was not the case found by [54], where the linear twist distribution resulted

in better performance for the NREL S814 blade with blade specifications defined by Barltrop [58]. Nonetheless, a better performance is generally obtained when feathering is more aggressive (from linear to hyperbolic trend). This indicates that blades having the base twist distribution were considered for round 2.

The hyperbolic chord distribution resulted in a generally lower $C_P$. The opposite was observed for the elliptical chord distribution, although the alteration also increased $C_T$. Both cases altered the solidity, which influences the blade loads, i.e., the hyperbolic chord distribution reduces the solidity, resulting in lower loads, while the opposite occurs for the elliptical chord distribution.

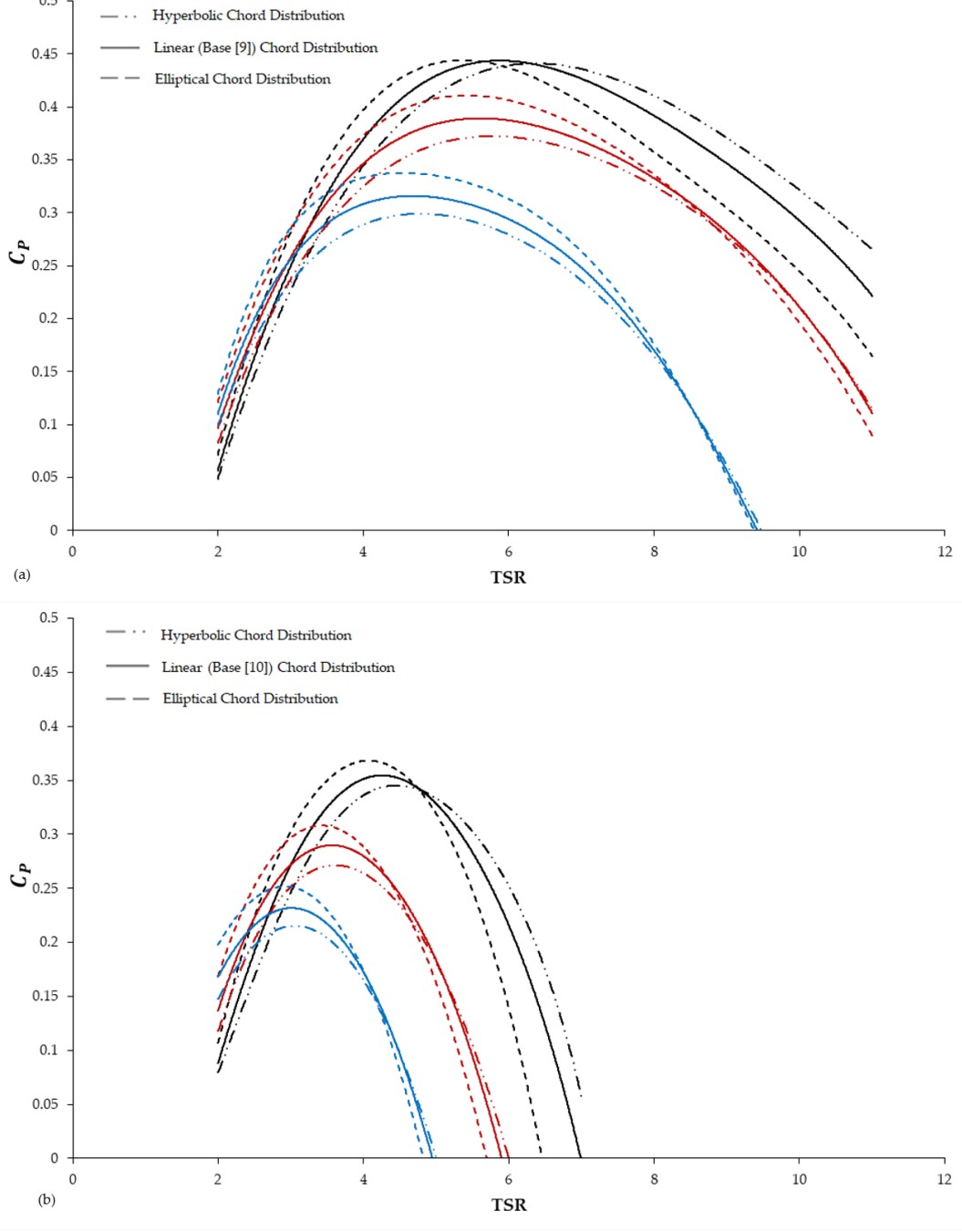

**Figure 7.** $C_P$ characterisation for Round 1 blades: (**a**) NACA and (**b**) NREL. Values are clustered into elliptical (blue), linear (red), and hyperbolic or base (black) twist distributions.

**Table 4.** Round 1: Averaged maximum $C_P$ and tip-speed ratio (TSR) locations according to twist distribution.

| Blade Twist Distribution | | Averaged Maximum $C_P$ | Averaged TSR Location |
|---|---|---|---|
| NACA | Hyperbolic(Base) | 0.436 | 5.92 |
| | Linear | 0.386 | 6.17 |
| | Elliptical | 0.316 | 5.17 |
| NREL | Hyperbolic(Base) | 0.354 | 4.25 |
| | Linear | 0.287 | 3.58 |
| | Elliptical | 0.233 | 3 |

### 3.3. Pool Round 2—Steady-State High-TSR Optimisation

As expected, the chord length reduction pushed the TSR location of the maximum $C_P$ towards higher TSRs (Figure 8). Near maximum performance remained similar for both original and low-solidity variations; a formatted table serves as a better illustration to show movement within the TSR range. For the NACA blades, the maximum $C_P$s were pushed by 0.75, 1.25, and 1.5 towards higher TSRs for the hyperbolic, base, and elliptical distribution trends, respectively. This shows a decrease in the resulting push towards higher TSRs as the unreduced chord length variation solidity decreases, i.e., $\Delta_{TSR,elliptical} > \Delta_{TSR,base} > \Delta_{TSR,hyperbolic}$; $\sigma_{elliptical} > \sigma_{base} > \sigma_{hyperbolic}$. The same is not true for the NREL blades as the degree of change remained constant for all distribution trends. The difference in behaviour may be specific to each aerofoil, but it may also be due to the difference in the overall solidity of the unreduced chord length variations. The latter would imply that there is a solidity wherein the increase in TSR location of the maximum $C_P$ suddenly shoots up until reaching a point where any decrease in solidity would have a marginal effect.

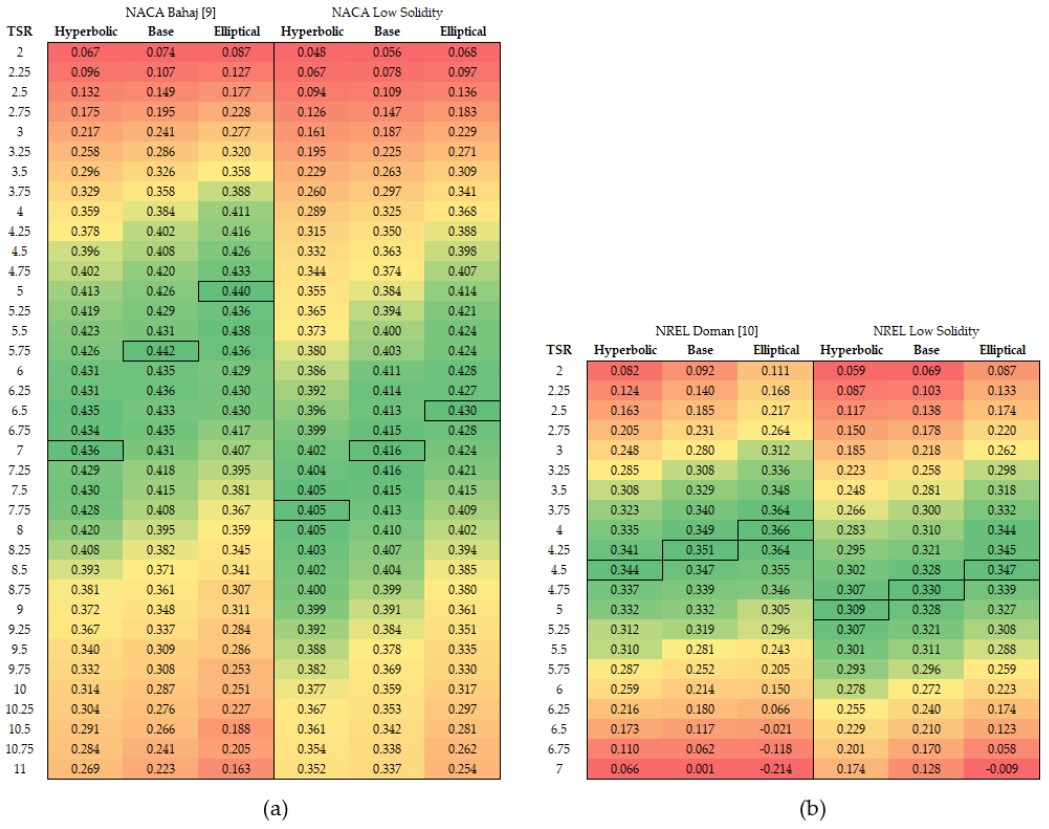

**Figure 8.** Locations of maximum $C_P$ (marked out values) for the (**a**) NACA and (**b**) NREL blades. TSR locations are pushed towards higher regions.

Maximum $C_P$ was found to be lower for both low-solidity aerofoil-specific blades compared to their respective unreduced chord length variations, with a 7.03% decrease for the NREL blades and a 5.02% decrease for the NACA blades. This is expected since a reduction in chord length leads to a decrease in the blade surface area that can interact with the fluid flow to extract energy. Table 5 shows the maximum $C_P$ and the corresponding TSR location for both the low-solidity and unreduced chord length variations of each blade.

**Table 5.** Maximum $C_P$ and corresponding TSR location for low-solidity blades.

| Blade Chord Distribution | | Maximum $C_P$ | TSR Location |
|---|---|---|---|
| NACA | Hyperbolic(Base) | 0.405 | 7.75 |
| | Linear | 0.416 | 7 |
| | Elliptical | 0.430 | 6.5 |
| NREL | Hyperbolic(Base) | 0.309 | 5 |
| | Linear | 0.330 | 4.75 |
| | Elliptical | 0.347 | 4.5 |

### 3.3.1. NREL S814 Blade

Similar to the results in round 1, the elliptical chord distribution resulted in the highest maximum $C_P$ among all the three low-solidity NREL blades, while the hyperbolic chord distribution resulted in the lowest maximum $C_P$. However, the difference in $C_P$ between the low-solidity base chord distribution and the low-solidity elliptical chord distribution was minimal ($C_{P,base} = 0.330$ vs. $C_{P,elliptical} = 0.347$). This translates to only a 5.07% increase in performance for an 18.7% increase in solidity.

The TSR location of the maximum $C_P$ was highest for the hyperbolic chord distribution with $TSR_{maxCp} = 5$. This means that no combination among the three blades dominates performance according to the set objectives, and MODM must be used if conditions are satisfied. Table 6 then shows the MODM result.

The base chord distribution was selected as the best low-solidity variation for the NREL blade. This variation generates 11.9% lower power output for both site cases under uniform flow.

**Table 6.** Weighted MODM Matrix.

| Blade Aerofoil | Objective | Weight | Scalarised Values | | |
|---|---|---|---|---|---|
| | | | Hyperbolic | Linear (Base) | Elliptical |
| NACA | Maximum $C_P$ | 0.5 | 0.941 | 0.903 | 1 |
| | TSR Location | 0.5 | 1 | 0.903 | 0.838 |
| | **Weighted Total** | **1** | **0.970** | **0.935** | **0.919** |
| NREL | Maximum $C_P$ | 0.5 | 0.890 | 0.952 | 1 |
| | TSR Location | 0.5 | 1 | 0.950 | 0.9 |
| | **Weighted Total** | **1** | **0.945** | **0.951** | **0.95** |

### 3.3.2. NACA 638xx Blade

Trends in maximum $C_P$ for the NACA blade are similar to the trends for the NREL blade, although the increase obtained by adopting an elliptical chord distribution was only at 3.46% ($C_{P,base} = 0.416$ vs. $C_{P,elliptical} = 0.430$). The significant change was observed in the TSR location of the maximum $C_P$.

The degree of change for the TSR location for the NACA blades was much larger, with a deviation that is almost 3 times greater compared to the NREL ($stdev_{NACA} = 0.629 > stdev_{NREL} = 0.25$). The higher degree of change may be explained by the already lower solidity of the original NACA blade, i.e., pushing the TSR location of the maximum $C_P$ is more pronounced as solidity goes lower. Due to this, the MODM is dominated by the TSR location, as shown in Tables 5 and 6.

The hyperbolic chord distribution was the best low-solidity variation for the NACA blade and was selected for further analysis. The maximum $C_P$ of 0.405 is comparable to the maximum $C_P$ produced by the second-best blade but the TSR location of 7.75 may lead to greater cost reduction due to less torque requirements. This variation generated 8.29% lower power output compared to the original geometry for both site cases under uniform flow.

### 3.4. Quasi-Unsteady Loads

Steady-state performance was only used for filtering blade variations, as discussed in Section 2.4.1 Comparison under profiled flow was performed to give a more realistic evaluation of performance between the original blades and the selected low-solidity variations. The power and torque for each blade were compared to substantiate the possible savings in generator and power take-off mechanisms, e.g., similar amount of power at reduced torque requirements. Each 10-s quasi-unsteady simulation was completed within 15 min for a time step of 0.1 s.

### 3.4.1. Profiled Flow Performance

Tables 7 and 8 show the torque and power of the two reduced chord length blades and the original aerofoil-specific blades. Under profiled flow, the optimised high-TSR NACA blade produced 9.85% less power than the original aerofoil-specific blade, accompanied by a 33.14% reduction in torque. The large torque reduction in the low-solidity variation provides the opportunity for significant generator downsizing without sacrificing too much power.

**Table 7.** 4 m turbine performance for site case 1: Philippines, $U(z) = 0.64z^{1/3.5}, h = 20$ m.

| Blade Aerofoil | Variation | Power (kW) | | Torque under Profiled Flow (Nm) |
| --- | --- | --- | --- | --- |
| | | Uniform Flow | Profiled Flow | |
| NACA | Bahaj [30] | 0.745 | 0.399 | 216.60 |
| | Low-Solidity [1] | 0.683 | 0.359 | 144.82 |
| NREL | Doman [31] | 0.591 | 0.317 | 233.37 |
| | Low-Solidity | 0.521 | 0.299 | 196.58 |

[1] The low-solidity blade variation has a hyperbolic chord distribution and not from Bahaj [30].

**Table 8.** 4 m turbine performance for site case 2: Mexico, $U(z) = 1.136z^{1/2.991}, h = 18.5$ m.

| Blade Aerofoil | Variation | Power (kW) | | Torque under Profiled Flow (Nm) |
| --- | --- | --- | --- | --- |
| | | Uniform Flow | Profiled Flow | |
| NACA | Bahaj [30] | 4.166 | 1.983 | 607.28 |
| | Low-Solidity | 3.821 | 1.789 | 406.33 |
| NREL | Doman [31] | 3.307 | 1.502 | 622.04 |
| | Low-Solidity | 2.915 | 1.439 | 533.40 |

The low-solidity NREL experienced less power loss, at 5.00% reduction in output with $z$ 15.00% reduction in torque. This may still lead to downsizing and savings, although it is not expected to be as large as what can be realised from the optimised high-TSR NACA blade, due to the smaller reduction in torque.

As discussed, the average flow over the rotor plane drops upon the assumption of a profiled flow, resulting in power loss for all blade profiles. All blade variations experience similar relative power loss, at 42–47% for the Philippines case and 50–55% for the Mexico case. The larger percentage drop in the Mexico case is explained by the lower value of b ($b_{Mexico} = 2.991 < b_{Philippines} = 3.5$), which results in a larger drop in average flow over the rotor plane.

A difference was observed between the NREL and NACA blades in their respective sensitivity upon the assumption of a profiled flow. The optimised high-TSR NREL blade reduced the relative

performance difference between uniform and profiled flow performance (Table 9). The reverse was observed for the NACA blades, although the increase in performance loss was small (<1%). Nonetheless, the relatively similar drops in performance for both the low-solidity variations and the original blades indicate that no additional effects from the reduction of chord lengths occur when the turbines are operated under profiled flow conditions.

**Table 9.** Relative power output difference between uniform and profiled flow assumptions.

| Blade Aerofoil | Variation | Relative Power Output Difference [1], % | |
| --- | --- | --- | --- |
| | | Case 1: Philippines | Case 2: Mexico |
| NACA | Bahaj [30] | 4.17 | 1.98 |
| | Low-Solidity | 3.82 | 1.79 |
| NREL | Doman [31] | 3.31 | 1.50 |
| | Low-Solidity | 2.92 | 1.44 |

[1] Relative difference computed with reference to uniform flow assumption.

### 3.4.2. Individual Blade Load Variation

Blade load variation for both torque and thrust were lower in the low-solidity blades (Figure 9 and Table 10). For the Mexico case, thrust variations for the low-solidity blades decreased by 6.74% and 5.65% compared to the original variations for the NACA and NREL blades, respectively. Torque variations decrease by a larger amount, with 33.54% and 26.83% for the NACA and NREL blades, respectively.

The drop in thrust variation was much greater for the Philippines case, although this is mostly attributed to the lower load magnitudes resulting from the lower flow velocity i.e., relative differences are magnified even in small absolute differences. Thrust variations in the low-solidity blades decreased by 16.23% and 28.31% compared to the original variations for the NACA and NREL blades, respectively. Torque variation decreased by a larger amount, with 32.87% and 26.39% for the NACA and NREL blades, respectively.

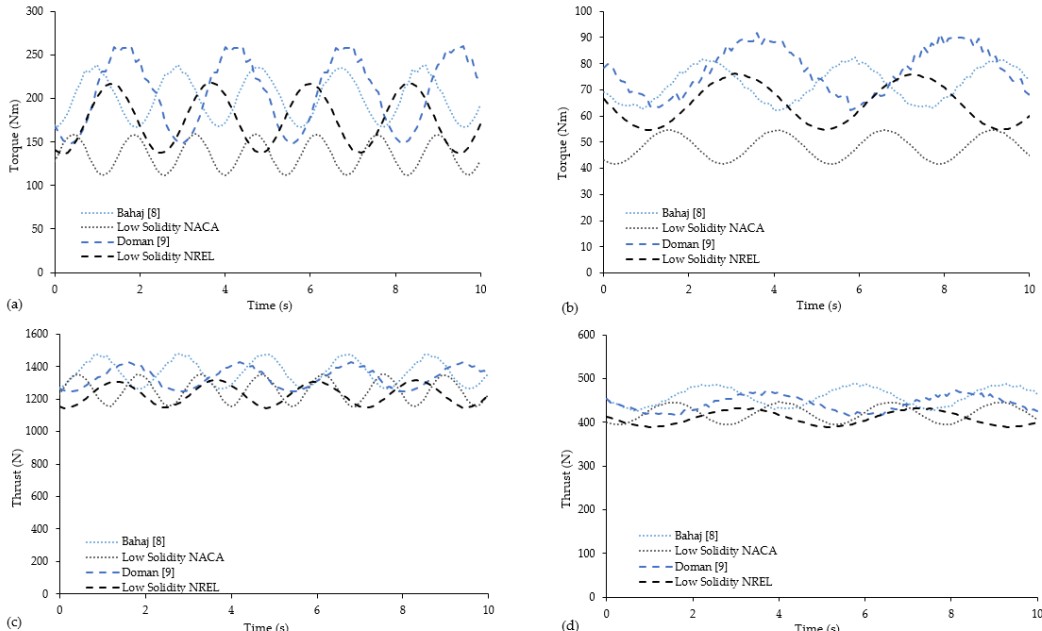

**Figure 9.** Profiled Flow simulation—(**Top**) Torque Fluctuations for the (**a**) Mexico case and (**b**) Philippines case; (**Bottom**) Thrust Fluctuation for the (**c**) Mexico case and (**d**) Philippines case.

The reduction of load variation favours the adoption of the low-solidity blades since this should lessen the effect of fatigue. However, the increased number of cyclic loading from the increased

rotational speed may offset the effect of the reduced fatigue loads, especially for the NACA blade; the reduced chord length NACA blades rotate 34.78% faster than the original blade.

**Table 10.** Load variations for each blade variation.

| Blade Aerofoil | Variation | Thrust Variation (N) | | Torque Variation (Nm) | |
|---|---|---|---|---|---|
| | | Philippines | Mexico | Philippines | Mexico |
| NACA | Bahaj [30] | 61.20 | 212.67 | 19.48 | 71.57 |
| | Low-Solidity | 51.26 | 198.34 | 13.08 | 47.56 |
| NREL | Doman [31] | 59.90 | 182.94 | 28.97 | 111.46 |
| | Low-Solidity | 42.94 | 172.60 | 21.33 | 81.55 |

### 3.4.3. Power Output

The simulated power outputs were generally low since the turbines were made to be only 4 m in diameter. However, it has been shown [25] that an increased number of smaller turbines in an array can still generate considerably large power. The perceived benefits in terms of lower load magnitudes and less fatigue may not be realised when larger diameter turbines are adopted, since larger diameters would lead to slower rotational speeds.

## 4. Conclusions and Future Work

A tidal turbine design process for less energetic currents has been presented. The best performing blade for each aerofoil-specific blade geometry was selected using the two set objectives of achieving a high maximum $C_P$ and high TSR location. The twist distribution was determined to have a more significant effect, and the best variation was found to be the base twist distributions as published by Bahaj and Doman.

The simulations show reasonable performance for high-TSR blades with a slight decrease in power. The hyperbolic chord distribution ($\mathbf{A} = 5$) was the best low-solidity variation for the NACA 638xx blade with a TSR location at 7.75 and a maximum $C_P$ of 0.405. The maximum $C_P$ value is roughly 8% lower than the maximum $C_P$ of the original Bahaj blade, but the 33% reduction in torque requirements and the significantly higher angular speed, $\Delta_{TSR} = 2$ can reduce the costs as smaller generators and direct-drive power take-off may be employed. The reduced chord-length variation of the original Doman blade gave the best performance for the low-solidity NREL blade. Similarly, the maximum $C_P$ of the low-solidity variation was lower by about 5% compared to the original blade. Cost reduction can be achieved, although it may not be as large as the NACA blade since the reduction in torque is only 15% with a lower increase in angular speed $\Delta_{TSR} = 0.5$.

The quasi-unsteady simulations show benefits for the high-TSR blades as load variations are minimised, with both low-solidity blades resulting in an averaged reduced torque and thrust variation of 30% and 6% decrease, respectively. However, additional analysis must be performed to quantify fatigue since the reduced loads are accompanied by an increase in the number of cyclic loads, with the NACA blades rotating 1.35 times faster than the original NACA blade.

The design methodology may be expanded to accommodate more blade variations and may be done programmatically with $\mathbf{A}$ as a parameter to change the chord and twist distributions. Additional parameters to alter blade geometry may also be added. Pool filtering with a conditional multi-objective decision model provides a simple decision method to select an optimised blade. Other objectives and constraints may be added to ensure that output blades are technically and economically feasible. These objectives and constraints may include cost-effectiveness, minimal deflection, cavitation, etc.

The blade-element momentum method is limited to quasi-unsteady analysis. Expansion to unsteady analysis with added mass methods should improve the accuracy of the simulations. The output of the single-parameter BEM resulted in highly erratic output for some of the simulations. This can be fixed by adding more constraints in convergence among others. Further increasing the

temporal resolution to 0.01 s may improve the accuracy for quasi-unsteady simulations, but additional mathematical corrections must be included as erratic values also occur at very high TSRs (TSR > 10) during steady-state simulations.

Wave–current interactions should also be considered. The current method of computing velocities at every time step already allows for the incorporation of changing velocities from wave–current interactions. The azimuthal coordinate system allows for vertical velocity contributions. Additional data on wave-induced velocities are needed to identify conditions in less energetic currents.

**Author Contributions:** J.I.E. and C.J. conceptualised and formulated the hypothesis. J.I.E. developed the blade design methodology, implemented BEM in python, and performed calculations. S.O.-S. provided datasets for Mexico. All authors discussed the anaylsis, verified the results, and contributed to writing and editing the original draft.

**Funding:** J.I.E. is funded to undertake research under the Foreign PhD Scholarship program of the Department of Science and Technology—Engineering Research and Development for Technology and the Foreign PhD Fellowship of the University of the Philippines Diliman.

**Acknowledgments:** The data used in this paper was obtained through Newton Fund Institutional Links grant IL5 332324562, CONACYT-SENER-Fondo de Sustentabilidad Energetica-Institutional Links grant IL5 291380, and OceanPixel Ltd. Aerodynamic characteristics of the NACA 638xx foils were obtained by Gavin Lavery as part of a CFD characterisation on multiple aerofoils.

**Conflicts of Interest:** The authors declare no conflict of interest.

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
