# Peer review of "Design of a Horizontal Axis Tidal Turbine for Less Energetic Current Velocity Profiles"

_jmse, doi:10.3390/jmse7070197_

Round 1
Reviewer 1 Report
Comments and Suggestions for Authors
The manuscript is an investigation about the design of a horizontal axis tidal turbine for less energetic current velocity profiles. This subject is particularly important due to the innovation and the hopeful results and the increasing market of tidal energy.
Please, check the whole manuscript, there are many typos and misprints. Please be more consistent in your writing and format style and apply the journal template correctly to the manuscript. Typos and grammar errors sprinkled over the paper
The abstract is extensive and well written; however some quantification of the results is lack.
INTRODUCTION: A deep review is not acutely described. The introduction is well written and structured but very short. Additionally I missed more JCR indexed paper. Please, state more clear the objectives and the necessity of the current study. The state of the art is not described in an exhaustive way.
The subsections of section 2 Methods are nicely described and organized.
Results are nice. Please explain in a better way the tables of Figure 8
When talking about BEM and power output calculation I recommend to the authors to read the following paper: https://www.mdpi.com/1996-1073/10/6/742 (https://doi.org/10.3390/en10060742)
Section 4: conclusions
The conclusions are not clearly described and very short. Please, describe in more detail all the conclusions and findings of the current study. The conclusions have to be based on the current results and acutely described and compared. Please, quantify the results and the comparisons as well.
Author Response
Response to Reviewer 1 Comments
Point 1: Please, check the whole manuscript, there are many typos and misprints. Please be more consistent in your writing and format style and apply the journal template correctly to the manuscript. Typos and grammar errors sprinkled over the paper.
Response 1:
Proofread the paper again to correct spelling and grammar.
I have used the MDPI LaTeX journal template for the original manuscript and just inputted text. The same LaTeX template was used for the figures and tables. This should ensure that the format is followed. The only section that veers away from the specified format is 2.4.1 and the reason is stated below.
The methodology section 2.4.1 has three more subsections in bold. I opted to write the subheadings in bold-face as the nested sections in the provided MDPI LaTeX template only allows up to a subsubsection. I can make the format 2.4.1.x for a subsubsubsection but I think bold-faced subheading is better since they are similar procedures applied to different blade geometries that are analysed in succession. I can also not include subheadings for Round 1 and 2 but opted for the inclusion of such to make it easier to read.
Point 2: The abstract is extensive and well written; however some quantification of the results is lack.
Response 2: Added quantifications for power drops, rotational speed increase, and torque drops (in blue).
Point 3: A deep review is not acutely described. The introduction is well written and structured but very short. Additionally I missed more JCR indexed paper. Please, state more clear the objectives and the necessity of the current study. The state of the art is not described in an exhaustive way.
Response 3:
Devices – added discussion on current technologies (in magenta/pink)
Necessity – added more references for less energetic sites (in violet). This works along side the review for current technology. Added statement on design that may expand market (in red)
Objectives – added another paragraph at the end for specific objectives. (in blue)
Point 4: Results are nice. Please explain in a better way the tables of Figure 8
Response 4:
Section 3.3 – Moved the footnote of Figure 8 to body (in red). Described the figure in more detail and implications (in blue).
Point 5: When talking about BEM and power output calculation I recommend to the authors to read the following paper: https://www.mdpi.com/1996-1073/10/6/742 (https://doi.org/10.3390/en10060742)
Response 5: The recommended paper has separate sections for BEM calculation and power calculation.
Section 2.2.1 – Added discussion for BEM with a single parameter (in blue).
Section 2.2.2 – Added clarification for the unsteady condition (in blue). The unsteady condition applies to each blade as it moves through the water column and thus, experiences different inflow conditions due to a profiled flow. The recommended paper has a calculation for the average power of the whole rotor for winds with temporal variations. This is not the same for the quasi-unsteady case investigated as the profiled flow still assumes that the inflow over the whole rotor (not the blade) is still constant with respect to time i.e. U(z) is constant at any time t. Power are not calculated outside BEM, added a clarification that it is directly calculated from Cp or the combination of torque and omega (Section 2.2.1, in blue).
Point 6: The conclusions are not clearly described and very short. Please, describe in more detail all the conclusions and findings of the current study. The conclusions have to be based on the current results and acutely described and compared. Please, quantify the results and the comparisons as well.
Response 6: Added quantifications (in blue) to make the existing text regarding the findings clearer (in red). I have opted for a joint conclusion and future work section so some text not found in the current study are included.

Reviewer 2 Report
General Comments:
The manuscript entitled “Design of a Horizontal Axis Tidal Turbine for Less Energetic Current Velocity Profiles” by Job Immanuel Encarnacion, et al. presents a design methodology for low-solidity high tip-speed ratio turbines aimed to operate at less energetic flows. The topic is interesting to the journal, it would find readers interested in tidal energy. The manuscript is well written and structured, and the methodology is adequate. Therefore, it should be accepted for publication.
There are some minor issues that need to be addressed before this paper can be accepted for publication.
Introduction
Line 33. In order to enrich your work, include a couple of references of tidal energy sites around the world with flow velocities below 2 m/s, e.g. São Marcos Bay, Brazil (https://doi.org/10.1007/s4072), Ria Formosa, Portugal (https://doi.org/10.1016/j.energy.2018.06.034)
Line 49. Check units of rotor diameter.
Methods
End of paragraph 2. “These distributions serve as an initial case study on their performance under less energetic flow l(r) is then applied as follows”. Revise sentence, a comma may be missing between “flow” and “l(r)”.
Line 114-117. Provides more details or references on how the Ansys Fluent simulations were performed.
Line 243. Check punctuation. “(Figure 3a) A load”.
Conclusion and Future Work
Add a note on the size and number of time-steps deemed, as well as the time taken to run the quasi-unsteady analysis.
Author Response
Response to Reviewer 2 Comments
Point 1: Line 33. In order to enrich your work, include a couple of references of tidal energy sites around the world with flow velocities below 2 m/s, e.g. São Marcos Bay, Brazil (https://doi.org/10.1007/s4072), Ria Formosa, Portugal (https://doi.org/10.1016/j.energy.2018.06.034)
Response 1: Added other areas with resource less than 2m/s in the Introduction (in blue)
Point 2: Line 49. Check units of rotor diameter.
Response 2: Noted and changed to ‘m’.
Point 3: End of paragraph 2. “These distributions serve as an initial case study on their performance under less energetic flow l(r) is then applied as follows”. Revise sentence, a comma may be missing between “flow” and “l(r)”
Response 3: Noted and added a period between flow and l(r)
Point 4: Line 114-117. Provides more details or references on how the Ansys Fluent simulations were performed.
Response 4:
Section 2.2.1 – Added discussion on CFD setup (in blue) and reorganized to have one separate paragraph the setup.
Point 5: Line 243. Check punctuation. “(Figure 3a) A load”.
Response 5: Noted and added period in between ‘)’ and ‘A’.
Point 6: Add a note on the size and number of time-steps deemed, as well as the time taken to run the quasi-unsteady analysis.
Response 6:
Section 3.4 – Added statements on the time steps and the time it took for each analysis. (in magenta)
Conclusion – changed transient to erratic as some of the erratic values also appeared in the steady-state simulations although they are only limited to the very high end of the TSR spectrum (TSR > 10). Also added possible solutions: increase temporal resolution or add more mathematical constraints. (in blue)

Round 2
Reviewer 1 Report
all my requirements have been succesfylly addressed